# Chemical Characterization and Antimicrobial Properties of the Hydroalcoholic Solution of *Echinacea purpurea* (L.) Moench. and Propolis from Northern Italy

**DOI:** 10.3390/molecules28031380

**Published:** 2023-02-01

**Authors:** Stefania Garzoli, Francesca Maggio, Vittorio Vinciguerra, Chiara Rossi, Matthew Gavino Donadu, Annalisa Serio

**Affiliations:** 1Department of Drug Chemistry and Technology, Sapienza University, 00185 Rome, Italy; 2Department of Bioscience and Technology for Food, Agriculture and Environment, University of Teramo, 64100 Teramo, Italy; 3Department for Innovation in Biological Systems, Food and Forestry, University of Tuscia, 01100 Viterbo, Italy; 4Hospital Pharmacy, Azienda Ospedaliero Universitaria di Sassari, 07100 Sassari, Italy; 5Department of Biomedical Sciences, University of Sassari, 07100 Sassari, Italy

**Keywords:** GC-MS, volatile compounds, bioactive compounds, antimicrobial activity, derivatives

## Abstract

In this study, for the first time, the chemical composition of *Echinacea purpurea* (L.) Moench. and propolis (EAP) hydroalcoholic solution from the Trentino Alto Adige region of northern Italy was investigated by using SPME-GC-MS to describe the volatile content and GC-MS after silylation to detect the non-volatile compounds in the extractable organic matter. The antimicrobial activity of EAP hydroalcoholic solution was evaluated by Minimum Inhibitory Concentration (MIC) determination on 13 type strains, food and clinical isolates. Time Kill Kinetics (TKK) assays and the determination on swimming and swarming motility for 48 h gave more details on the mode of action of EAP solution. The results highlighted the presence of some terpenes and a large number of compounds belonging to different chemical classes. Among these, sugars and organic acids excelled. The EAP hydroalcoholic solution exhibited a strong antimicrobial activity in terms of MIC, with a clear decrease in the cellular load after 48 h. However, the bacterial motility may not be affected by the EAP treatment, displaying a dynamic swarming and swimming motility capacity over time. Given the complexity of chemical profile and the strong antimicrobial effectiveness, the EAP hydroalcoholic solution can be considered a source of bioactive molecules, deserving further investigation for the versatility of application.

## 1. Introduction

Since ancient times, the conventional treatment of diseases and the modulation of pain has been associated with natural product supplementation. The paleontological studies suggest that the use of plants as medicines can be traced back at least 60,000 years [1].

The molecular complexity distinguishes the natural compounds for the mechanisms of action and versatility of application. Therefore, they can aid in developing today’s new drugs, as well as dietary supplements, which are designed to ease pathological chronic cases such as chronic kidney disease [2], osteoarthritis [3], diabetes, inflammatory bowel, Parkinson’s or Alzheimer’s disease [4]. Currently, careful attention is paid to natural compounds with wide-ranging mechanisms of action. Belonging to these classes, propolis and *Echinacea* are in the spotlight. Propolis is a natural resinous substance collected from flowers and leaves by *Apis mellifera* L. and exploited as a bioproduct to be used as a glue for beehives [5]. The chemical composition of propolis is associated with the geographical region of origin; however, the known components are grouped into chemical classes such as terpenoids, alcohols, esters, aromatic acids and volatile oils [6]. The therapeutic importance of propolis is based on its versatile effects. The antibacterial activity causes physical alterations of the bacterial cell membrane, the deformation and leakage of cell components, as well as the downregulation of biofilm-associated genes [7]. Widespread attention has been received for the anticancer properties, mainly directed to altering cell apoptosis and reducing the chronic inflammatory disorders [6]. Furthermore, antiviral activity, but also anti-inflammatory, antioxidant and hepatoprotective properties, have been recognized [8].

Just as important as propolis is *Echinacea purpurea* (L.) Moench. The genus *Echinacea* belongs to the Asteraceae family, consisting of 11 taxa of herbaceous and perennial flowering plants [9]. It is a medicinal herb native to North America and, nowadays, its use is increasing exponentially, especially as a crude drug. The health properties include antiviral, antibacterial and immunomodulatory activities, but also cells’ proliferative effects, traditionally associated with wound healing [10]. The presence of a polysaccharide fraction (echinacin B) by producing a hyaluronic acid–polysaccharide complex would stimulate wound healing, leading to the inhibition of hyaluronidase and promoting the fibroblasts’ growth [9]. Moreover, the stimulation of monocytes and natural killer cells would induce the first line of immune defense in the body against infections [9]. Available literature provides a complete picture of propolis and *E. purpurea* properties as diet supplementation, but there is a lack of knowledge on the pair’s combination. Hence, in this work, for the first time, the hydroalcoholic solution of *Echinacea purpurea* (L.) Moench. and propolis (EAP), from the Trentino-Alto Adige region of Italy, was chemically characterized by SPME-GC-MS to describe the volatile fraction and by GC-MS of the derivative extract to detect the non-volatile organic molecules characterizing the solution. Furthermore, the antimicrobial effectiveness was evaluated in terms of Minimal Inhibitory Concentration, growth inhibition over time (monitoring the TKK) and the influence on swimming and swarming motility. The antimicrobial analyses were performed on a heterogeneous set of clinical, environmental, food and type strains.

## 2. Results

### 2.1. Chemical Volatile Composition

By SPME-chromatographic analyses carried out on the untreated matrix, six volatile components were detected and identified (Table 1). All compounds belonged to the terpene family, and the monoterpenic content exceeded the sesquiterpenic one. Limonene was the major component followed by *p*-cymene, *δ*-cadinene and *α*-pinene. *α*-Farnesene and cis-muurola 3,5-diene represented the minor components, with similar percentage values.

### 2.2. Chemical Composition of the Hydroalcoholic Solution (EAP) after Derivatization

Direct injection analyses of the silylated extract allowed the identification of twenty-five compounds belonging to different chemical including organic acids, phenolic acids, fatty acids, sugars and alcohols (Table 2). Sugars represented the highest number of components found in EAP and, among them, D-fructofuranose followed by D-arabinopyranose, D-mannose and D-tagatofuranose were the major molecules. Nine different phenolic acids were identified; *p*-coumaric acid and isoferulic acid represented the components with higher percentage values. Three fatty acids, such as palmitic, lauric acid and α-linolenic, and the phenolic one, caffeic acid, were also detected. Lastly, a significative amount of vanillin was measured. 

### 2.3. MICs and MBCs Determination

As expected, the strain’s biodiversity reflected different responses to treatment. The MICs results for the bacterial strains under this study comprised between <300 and 700 µL/mL (Table 3), with the same values after 24 and 48 h of incubation, thus confirming the stability of the antimicrobial effect. EAP solution displayed a strong antimicrobial activity in terms of MICs. As shown in Table 3, low MIC values were mainly detected, and, in detail, <300 (58% of the strains) and 400 µL/mL (38% of the strains). Only *P. aeruginosa* ATCC 27853 (strains S7) showed growth capability in the presence of higher concentrations of EAP solution, reaching a MIC value of 700 µL/mL. The MBCs’ results (Table 3) confirmed the same trend detected for the MICs after 24 and 48 h, thus indicating that the inhibitory effect was due to the inactivation of the strains.

For the subsequent analyses, one representative strain of the whole set was chosen, and in detail, *L. monocytogenes* ATCC 7644 (strain S5). 

### 2.4. The Kinetics of Inactivation after EAP Solution Exposition

TKK (Figure 1) assay, monitored over 48 h of incubation, evidenced that EAP solution significantly reduced the cells’ load of *L. monocytogenes* ATCC 7644 (strain S5). The untreated cells showed an exponential load increase that, starting from 5.0 Log CFU/mL, reached a cellular load of 7.8 Log CFU/mL after 48 h of incubation at 37 °C. Since the first hour of exposition to 350 and 400 µL/mL of EAP extract, a slight, although not statistically significant, load decrease of 0.6 and 0.9 UFC/mL was noticed. Only after 24 h of EAP treatments, a drastic and significant cellular reduction below the detection limit of 2.0 Log CFU/mL was observed, maintaining the critical condition until the end of the experiment.

### 2.5. Influence of EAP Solution on Cellular Motility

As shown in Table 4, the S5 untreated cells showed an inert swimming and swarming behavior, suggesting a limited cellular motility. However, in the presence of 350 and 400 µL/mL of EAP solution, the surface colonization was observed. The swimming and swarming motility was positively influenced by the treatment, increasing the migration halo over time. After 48 h in presence of 350 µL/mL of EAP solution, the surface colonization reached 7.0 cm for swimming and 7.8 cm for swarming motility. Similarly, the exposition to 400 µL/mL of EAP caused a migration of about 7.9 cm and 7.0 cm for swimming and swarming motility, respectively.

## 3. Discussion

In this work, a hydroalcoholic solution of *Echinacea purpurea* and propolis was subjected to chemical analysis by SPME-GC-MS to measure the volatile content and by GC-MS of the silylated derivatives to measure the non-volatile one. To the best of our knowledge, this is the first work dealing with the chemical composition of the hydroalcoholic solution of *E. purpurea* and propolis. Until now, the only study regarding this matrix reported the determination of the total content of flavonoids and polyphenols by using the Folin–Ciocalteu and colorimetric assays and the measurement of the antioxidant activity [11].

Some previous works have been conducted on multiple species of *Echinacea* using different analytical methodologies and, in agreement with our data, phenolic acids represented the more abundant fraction. The content of total phenolics and caffeic acid derivatives in aerial parts and roots of *E. purpurea* grown in China was investigated by HPLC and colorimetric analysis. The results showed that the contents were generally higher in fresh than in dried raw material [12]. Phenolic compounds were determined by TOF-LC/MS in methanolic and aqueous extracts obtained from leaves and flowers of *E. purpurea* and *pallida*. Cichoric acid was the most abundant in the methanolic extracts of both species, but not in the aqueous ones, where caffeic acid prevailed, although in small quantities [13]. Ethanol extracts of *E. angustifolia* DC roots from different natural geographic areas were investigated by HPLC. The results highlighted a trend of caffeic acid and its derivatives, which contributed to differentiating the investigated chemotypes [14].

Regarding propolis, its chemical composition varies according to different factors such as the season, the climate, the hive and the geographical area of the honey [15,16]. Consequently, multiple compositional profiles have been described. In general, the main components were phenolic compounds, especially cinnamic acid derivatives and flavonoids, but also amino acids and amines [17,18]. By LC-MS technique, the phenolic compounds content of 19 propolis samples from 17 beekeepers of 5 Finnish provenances, was measured. The compounds found in the largest quantities were methyl-naringenin and caffeic acid phenethyl ester [19]. Bankova et al. [20] investigated ten propolis samples from Bulgaria, Italy and Switzerland by GC-MS. The results showed how the compositions of the one of Swiss origin, rich in phenolic glycerides, and the Sicilian one, rich in diterpenic acids, differed considerably from the others [20]. However, despite this variability in phytochemical composition, propolis has general pharmacological value [21].

Concerning the antimicrobial properties, the EAP solution displayed a strong effectiveness, which affected the bacterial growth capability over time. Moreover, it is worthy of consideration that the antimicrobial effect was exerted not only on type strains and food isolates but also on pathogenic isolates. In addition, Gram negative bacteria generally show higher resistance to antimicrobial compounds because of the composition of their cell wall, but in our case, no differences in terms of sensitivity to EAP were observed, with the only exception of *Pseudomonas aeruginosa* ATCC 27853 (strain S7). The molecular complex that characterizes the EAP solution could exert the relevant bioactivity. Przybyłek et al. [22] argued that the propolis acts on the permeability of the cellular membrane with disruption of membrane potential, thanks to hydrolytic enzymes production, which impairs the structure of the membrane [22]. On the other hand, several formulations of *Echinacea* exhibit a promising antibacterial potential against different pathogenic bacteria, including those that provoke respiratory diseases [9]. The potent and selective antiviral and antibacterial properties of *Echinacea* are due to the overturn of the proinflammatory cytokine stimulation, independently of the bacteria or virus that is causing the infection [23]. The immunomodulatory activities of *Echinacea* have been attributed to the presence of glycoproteins, alkylamides and polysaccharides [23]. Moreover, the pronounced inhibitory effect of EAP may be attributable to its abundance in phenolic terpenes, such as limonene (42.4%) and *p*-cymene (21.5%, Table 1), which display an arsenal of biological mechanisms of action as antioxidant, anti-inflammatory, anticancer and antimicrobial activities. Limonene and *p*-cymene affect the cytoplasmatic membrane with irreversible destruction of the lipids bilayer, bringing structural expansion and perturbation of membrane potential [24,25]. Furthermore, the high concentration of *p*-coumaric (39.8%) and caffeic acids (4.6%) can emphasize the antibacterial activity. The first could provoke irreversible permeability changes in the cell membrane, causing cells to lose the ability to maintain cytoplasm macromolecules and binding to DNA to inhibit cellular functions [26]. The caffeic acid has recently been discovered to generate reactive oxygen species upon photoirradiation by oxidation and to exert bactericidal activities [27]. However, in this study the EAP solution did not inhibit the bacterial motility capacity. *L. monocytogenes* ATCC 7644 cells under treatment with 350 and 400 µL/mL displayed a more dynamic swarming and swimming motility over time, if compared to control conditions. This behavior could be a response to the stressing condition. The marked motility could be a bacterial defense mechanism, due to exposure to a hostile environment. In fact, previous studies have demonstrated that a sufficient concentration of motile cells can spontaneously form high-density clusters, affecting the collective tolerance to antimicrobial treatments that are lethal to planktonic cells [28,29].

The antimicrobial activity exerted by the EAP solution on microorganisms with different origins and biological characteristics underlined the promising versatility of the extract, as a dietary supplement, food biopreservative or as a healing ointment. However, a comparison with similar natural products is not possible because of the limited availability of literature on such extracts. Therefore, the EAP solution deserves further attention to deepen the knowledge of its bioactivity and its possible *in situ* application. 

## 4. Materials and Methods

### 4.1. Plant Materials

A hydroalcoholic solution of *Echinacea purpurea* (L.) Moench. and propolis (Trade Name: Echinacea-Propolis•Propoli; Identification Number: 8033745021957; Batch: 122502; Date of Manufacture: August 2019) was directly provided by Bergila GmbH Srl (Falzes/Issengo-Bolzano). The commercial hydroalcoholic solution was prepared starting from 50 mg of *E. purpurea* aerial parts (fresh flowers and roots) and 44 mg of propolis. *E. purpurea* aerial parts were collected in Falzes, Bolzano (Trentino-Alto Adige, Italy). *E. purpurea* roots and flowers were macerated with ethanol (at 70°) and left to infuse for 5–6 weeks. The propolis was cut and pulverized and macerated with organic ethanol (at 85°) for 5–7 weeks. Subsequently, the solutions were filtered and combined, and the graduation of the final tincture was reduced to 50–55° with distilled water.

Propolis deposited by the bees on a perforated stainless-steel grid was collected by the beekeeper by scraping. A smaller portion was also derived from the scraping of propolis from the hives and supers. The hives used were made of wood painted with boiled linseed oil and natural pigments avoiding the use of plastic. Once collected, the propolis was stored in the dark in a cool and dry place.

### 4.2. Materials

For extraction and derivatization, acetone, pyridine and bis-(trimethylsilyl) trifluoroacetamide (BSTFA) were purchased from Sigma-Aldrich (Steinheim, Germany).

### 4.3. SPME Sampling

To describe the volatile chemical profile of the hydroalcoholic solution, the SPME sampling technique was used. About 2 mL of the solution were placed inside a 7 mL glass vial with PTFE-coated silicone septum. To collect the volatiles in the adsorption phase, a SPME device from Supelco (Bellefonte, PA, USA) with 1 cm fiber coated with 50/30 μm DVB/CAR/PDMS (divinylbenzene/carboxen/polydimethylsiloxane) was used. Before use, the fiber was conditioned at 270 °C for 30 min. After achieving equilibration, obtained by heating to a suitable temperature and time, the fiber was exposed to the headspace of the samples for 25 min at 40 °C to capture and concentrate the components. Lastly, the SPME fiber was inserted in the GC injector maintained at 250 °C in split mode for desorption of the compounds.

### 4.4. GC-MS Analysis

To investigate the headspace from the solution, the analysis was carried out on a Clarus 500 model Perkin Elmer (Waltham, MA, USA) gas chromatograph coupled with a single quadrupole mass spectrometer (Clarus 500 model Perkin Elmer) equipped with a FID (flame detector ionization). The chosen capillary column was a Varian Factor Four VF-1. The GC oven’s programmed temperature was set initially at 60 °C and then increased to 220 °C at 6°/min and finally held for 15 min. Helium was used as a carrier gas at a constant rate of 1 mL/min. MS detection was performed with electron ionization (EI) at 70 eV operating in the full-scan acquisition mode in the m/z range 40–500 amu. The identification of compounds was performed by the comparison of the MS-fragmentation pattern of the analytes with those of pure components stored in the Wiley 2.2 and Nist 02 mass spectra libraries database. Further, the Linear Retention Indices (LRIs) were calculated using a series of alkane standards (C_8_–C_25_
*n*-alkanes-Agilent). The obtained LRIs were compared with available retention data reported in the literature. The relative amounts of the components were expressed as a percent peak area relative to total peak area without the use of an internal standard and any factor correction. The analysis was carried out in triplicate.

### 4.5. GC-MS Analysis of the Solution after Derivatization

To describe the non-volatile content of the solution, a derivatization reaction was performed. For this purpose, 20 mL of the hydroalcoholic solution were dried under reduced pressure at 37 °C to obtain 43.0 mg of solid residue. The solid material was washed 3 times with 2.0 mL of acetone, and the extract combined and dried under reduced pressure at 30 °C to obtain 15.0 mg of residue.

Subsequently, 1 mg of extract was added to 300 µL of pyridine and 100 µL of bis-(trimethylsilyl) trifluoroacetamide (BSTFA) with heating at 60 °C for 30 min. One μL of the silylated sample was manually injected at 270 °C into the GC injector in the splitless mode. The analysis was performed using the same apparatus GC-FID/GC-MS and the same capillary column (Varian Factor Four VF-1). The oven temperature program was as follows: 60 °C then a gradient of 7 °C/min to 170 °C for 1.0 min and a gradient of 8 °C/min to 250 °C for 25 min. Mass spectra were acquired in an electron impact mode. The identification of compounds was based on the percentage of similarity plus comparison of mass spectra (MS) using software NIST data library, with the percentage of total ion chromatograms (TIC%). Relative percentages for quantification of the components were calculated by electronic integration of the GC-FID peak areas, and no response factors were calculated.

### 4.6. Bacterial Strains and Culture Conditions

A collection of 13 strains of different species ad origins (Table 5) was selected for this research. *Listeria monocytogenes*, *Pseudomonas fluorescens* and *Salmonella enterica* ser. Veneziana and Kasenyi strains belonged to the collection of the Department of Bioscience and Technology for Food, Agriculture and Environment of Teramo University (Teramo, Italy). On the other hand, the wild strains of *Enterococcus faecalis* and *Ent. faecium*, *Staphylococcus epidermidis c. oculistice* and *Candida albicans* were kindly provided by the clinical collection of the Otolaryngology Clinic, Department of Medical, Surgical and Experimental Sciences, University of Sassari (Sassari, Italy). The fresh cultures cultivated on Müeller–Hinton (MH) agar plates (Liofilchem, Roseto degli Abruzzi, Italy) were inoculated in MH broth (Liofilchem, Roseto degli Abruzzi, Italy) and incubated at 37 °C or 30 °C (only for *P. fluorescens* strains) for 18 h, to obtain an early stationary phase fresh culture. Then, the standardized inocula were obtained by measurement in Lambda bio 20 spectrophotometer and appropriately diluted to 10^7^ or 10^5^ CFU/mL, in relation to the kind of analysis.

### 4.7. Minimum Inhibitory Concentrations (MICs) and Minimum Bactericidal Concentrations (MBCs) Assay

MICs were determined starting from a different concentration of the EAP hydroalcoholic solution, comprised from 200 to 700 µL/mL. The MIC values were determined at 37 °C and 30 °C (for *P. fluorescens*) after 24 and 48 h. MICs were considered as the lowest concentration of the solution, where the absence of red discoloration of TTC (2,3,5-triphenyltetrazolium chloride, Sigma-Aldrich, Milan, Italy), previously added to MH broth in a ratio of 1 µL/mL, was detected. The MBCs were assessed from the MICs’ wells, through plating out onto MH agar plates after 24 and 48 h of incubation. The analyses were performed in three biological replicates.

### 4.8. Time Kill Kinetics (TKK) Assay

In agreement with CLSI Guidelines [30], the TKK assay was performed considering *L. monocytogenes* ATCC 7644 strain treated with EAP solution for 1, 24 and 48 h. Five-hundred microliters of standardized bacterial cells (10^5^ CFU/mL) were treated for 60 min at 37 °C with 350 and 400 µL/mL of EAP solution. The untreated cells were assessed with bacteria incubated in PBS 10 mM (pH 7.4) and without EAP treatment. Then, 100 µL of the treated and untreated cells were centrifuged at 13,000 rpm for 5 min, washed 3 times with PBS 10 mM (pH 7.4) and, then, the cells were enumerated through plate count on MH agar plates after 48 h of incubation at 37 °C. The experiment was performed in triplicate.

### 4.9. Cellular Motility in Presence of EAP Treatment

Swimming and swarming motility of *L. monocytogenes* ATCC 7644 strain were assessed in presence of treatment with 350 and 400 µL/mL of EAP solution. The analysis was performed as described by Rossi et al. [31]. Ten microliters of standardized cells at 10^5^ CFU/mL were spotted on swimming and swarming media, performed as reported in Rossi et al. [31], previously spread with 20 µL/mL of each EAP solution concentration. Following the incubation at 25 °C for 24 and 48 h, the swimming and swarming motilities were evaluated by measuring the distance (cm) of the cells from the inoculation point, expressed as diameter (cm). The assay was performed in triplicate.

### 4.10. Data Analysis

Experimental results were subjected to ANOVA statistical analysis performed through XLSTAT ver. 2017. Data obtained were subjected to pair comparison within the same group, employing the test of Dunnett’s (for TKK results) and Tukey’s (for motility data), determining the statistically significant differences between each group, with * *p* values < 0.05. 

## 5. Conclusions

In the present study, numerous chemical compounds characterizing the commercial hydroalcoholic solution of *E. purpurea* (L.) Moench. and propolis have been identified, thus demonstrating how this solution can be considered as a source of bioactive molecules. In addition, biological activities against the bacterial growth capability were revealed from the first hours of treatment, suggesting a dramatic stress condition of the cells.

In conclusion, the obtained data indicate that this natural product deserves to be further investigated to carry out composition–activity correlation studies and to evaluate a potential exploitation in food or clinical environments.

## Figures and Tables

**Figure 1 molecules-28-01380-f001:**
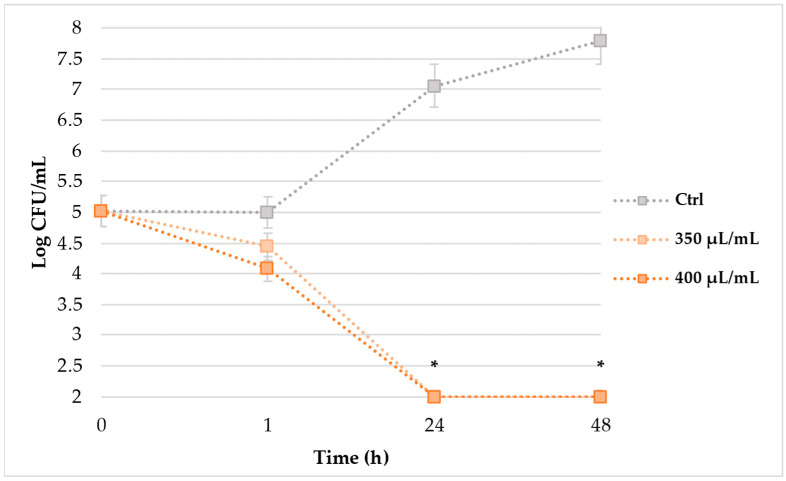
TKK results (Log CFU/mL) of S5 strain after 1, 24 and 48 h of exposition to 350 and 400 μL/mL of EAP solution. Asterisks mean statistically significant differences (* *p* < 0.05) between the cells treated with 350 and 400 µL/mL of EAP solution, compared to control cells.

**Table 1 molecules-28-01380-t001:** Chemical volatile composition (percentage mean value ± standard deviation) of EAP hydroalcoholic solution.

N°	COMPONENT ^1^	LRI ^2^	LRI ^3^	EAP
1	*α*-pinene	951	945	9.1 ± 0.05
2	*p*-cymene	1012	1016	19.0 ± 0.15
3	limonene	1018	1023	42.4 ± 0.21
4	*α*-farnesene	1487	1484	5.8
5	cis-muurola 3,5-diene	1451	1447	5.9
6	*δ*-cadinene	1530	*	17.8 ± 0.12
	SUM			100.0
	Monoterpenes			70.5
	Sesquiterpenes			29.5

^1^ The components are reported according to their elution order on the apolar column; ^2^ Linear Retention Indices measured on the apolar column; ^3^ Linear Retention indices from literature; * LRI not available; EAP: Percentage mean values of *E. purpurea* (L.) Moench and propolis solution components.

**Table 2 molecules-28-01380-t002:** Chemical composition (percentage values) of EAP after derivatization.

N°	COMPONENT	EAP (%) ^1^
**ORGANIC ACIDS**		
1	lactic acid	0.3
**PHENOLIC ACIDS**		
2	caffeic acid	4.6
3	vanillic acid	0.4
4	salicylic acid	0.3
5	cinnamic acid	1.0
6	3-hydroxycinnamic acid	4.8
7	3,4-dimethoxycinnamic acid	1.6
8	*p*-coumaric acid	39.8
9	ferulic acid	1.4
10	isoferulic acid	15.7
**FATTY ACIDS**		
11	lauric acid	0.6
12	palmitic acid	0.8
13	α-linolenic acid	0.3
**SUGARS**		
14	D-glucose	1.8
15	D-mannose	2.1
16	D-arabofuranose	0.1
17	D-tagatofuranose	2.0
18	L-sorbofuranose	1.1
19	D-ribofuranose	0.4
20	D-fructofuranose	5.7
21	α-arabinofuranoside	0.5
22	methyl-α-D-glucofuranoside	0.8
23	D-arabinopyranose	2.4
**OTHERS**		
24	vanillin	4.5
25	benzylcinnamate	0.7

^1^ Percentage values of the components of EAP hydroalcoholic solution after derivatization.

**Table 3 molecules-28-01380-t003:** MICs and MBCs after 24 and 48 h of EAP hydroalcoholic solution against the strains under this study.

	MICs (µL/mL)												
Time (h)	S1	S2	S3	S4	S5	S6	S7	S8	S9	S10	S11	S12	S13
24	<300	400	<300	<300	400	<300	700	<300	<300	400	<300	400	400
48	<300	400	<300	<300	400	<300	700	<300	<300	400	<300	400	400
	**MBCs (µL/mL)**												
**Time (h)**	**S1**	**S2**	**S3**	**S4**	**S5**	**S6**	**S7**	**S8**	**S9**	**S10**	**S11**	**S12**	**S13**
24	<300	400	<300	<300	400	<300	700	<300	<300	400	<300	400	400
48	<300	400	<300	<300	400	<300	700	<300	<300	400	<300	400	400

The results of MICs and MBCs were expressed as μL/mL.

**Table 4 molecules-28-01380-t004:** EAP solution influence on swimming and swarming motility at 25 °C in *L. monocytogenes* ATCC 7644 (strain S5).

Swimming				Swarming		
Time (h)	Ctrl	350 µL/mL	400 µL/mL	Ctrl	350 µL/mL	400 µL/mL
**24**	0.6 ^a^	4.3 ^b^	6.6 ^b^	0.5 ^a^	2.9 ^ab^	3.2 ^b^
**48**	0.6 ^a^	7.0 ^b^	7.9 ^b^	0.5 ^a^	7.8 ^b^	7.0 ^b^

Results are expressed as the diameter of the migration in cm. Ctrl: Control; 350 and 400 µL/mL of EAP solution. Different letters mean statistically significant differences (* *p* < 0.05) compared to the control at the same time of incubation.

**Table 5 molecules-28-01380-t005:** Bacterial strains under this study.

ID Code	Strain	Origin
S1	*C. albicans* 551 RM	Clinical
S2	*E. coli* ATCC 35218	Type
S3	*Ent. faecalis* 02/02/2017	Clinical
S4	*Ent. faecium* 02/02/2017	Clinical
S5	*L. monocytogenes* ATCC 7644	Type
S6	*L. monocytogenes* 641/6II	Cold smoked salmon
S7	*P. aeruginosa* ATCC 27853	Type
S8	*P. fluorescens* ATCC 13525	Type
S9	*P. fluorescens* 349.1	Dairy products
S10	*St. aureus* ATCC 43300	Type
S11	*St. epidermidis c. oculistice* 3	Clinical
S12	*S.* Kasenyi	Fresh and minimally processed fruits and vegetables
S13	*S.* Veneziana	Fresh and minimally processed fruits and vegetables

## Data Availability

Data available on request due to restrictions of privacy. The data presented in this study are available on request from the corresponding author.

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
