# Peer review of "Chemical Characterization and Antimicrobial Properties of the Hydroalcoholic Solution of Echinacea purpurea (L.) Moench. and Propolis from Northern Italy"

_molecules, 2023, doi:10.3390/molecules28031380_

Round 1

Reviewer 1 Report

4.2. Extraction process: Please, provide more details about the extraction process. After the second drying process, the 15.0 mg remaining were further diluted prior to the SPME sampling?

Revise the numbering of the sections: 4.2. was used for Extraction process and SPME sampling. 

SPME sampling: "About 2.0 mL of each extract..." Which extracts were evaluated here? Please, describe clearly the samples evaluated. 

4.6. MICs and MBCs assays: instead of using the acronyms here, please include the full name of the analyses.

TKK assay: please use the fill name of the analysis. 

Author Response

Reviewer 1

Dear Reviewer,

Thank you for the care used to revise our manuscript and for the positive observations. We have revised it according to the suggestions incoming from you and the other reviewer.

4.2. Extraction process: Please, provide more details about the extraction process. After the second drying process, the 15.0 mg remaining were further diluted prior to the SPME sampling?

We thank the reviewer for the observation. The analysis performed by SPME-GC-MS was carried out directly on the starting hydroalcoholic solution (2 mL). The second extraction process was carried out to remove the water content to then carry out the derivatization reaction.

To avoid any confusion and better clarify the sequence of the experimental steps, we thought it appropriate to move section 4.2 and integrate it with section 4.5.

Revise the numbering of the sections: 4.2. was used for Extraction process and SPME sampling.

We rearranged the sections as reported in the previous answer.

SPME sampling: "About 2.0 mL of each extract..." Which extracts were evaluated here? Please, describe clearly the samples evaluated.

To clarify that the SPME sampling technique was performed directly on the starting hydroalcoholic solution, the sentence has been changed.

4.6. MICs and MBCs assays: instead of using the acronyms here, please include the full name of the analyses.

TKK assay: please use the fill name of the analysis.

We thank you for the suggestions. The full names of the analyses were specified, in agreement to the advice.

Reviewer 2 Report

It is a work that leaves many doubts regarding the conditions for obtaining the hydroalcoholic solution of Echinacea purpurea (L.) Moench. and propolis that it is very important to know them in detail in order to carry out the replication and scientific validation of the results presented. In addition, the place of origin (from Northern Italy) is the place of the company, they do not ensure that it is where the plants and the propolis were produced.

 As mentioned in the manuscript, there are many factors that can influence the chemical composition of each of the two products used; therefore, with the commercial product they used (which does not describe any of its characteristics) the reproducibility of the experiment cannot be guaranteed.

They should have done the collection, handling and processing of the plant and obtaining the propolis, by themselves. Or ensure that the commercial production of the studied product will always be under the same characteristics, factors and conditions.

In addition, I make the following specific observations:

 L79-82: I suggest removing the numerical values that are in parentheses, since the values can be clearly seen in the nearby table. It's redundant. Apply the same criteria in writing the other results.

 L99-100: Table 2 header "COMPONENT1" has superscript 1 and is not described in table footer. Please put its meaning. In addition, it is important to explain the basis for the classification made for organic acids and phenolic acids. For example, why does caffeic acid classify it as a phenolic acid and p-coumaric acid classify it as an organic acid? when both are considered as organic compounds and classified as very similar hydroxycinnamic acids.

 L122-125: It is an error to ensure that a slight decrease in the bacterial load was noted from the first hour of exposure, when the statistical analyzes indicate that there was no significant difference (absence of asterisk) between the three treatments. Therefore, there is no decrease. Correct the wording.

 L125-127: Se desconoce si esa reducción fue causada por los compuestos en la solución de EAP o el diluyente (¿tal vez alcohol?) utilizado para el extracto. Por eso es importante que definan claramente todo el procesamiento del extracto.

 L129-131: Why are the concentrations different at the beginning of the caption (350 and 400 uL/mL) and at the end (300 and 400 uL/mL)? correct at which concentration the actual test was done (at 350 or 300 uL/mL).

 L141-146: Why do you mention in Table 4 a concentration of 300 uL/mL of EAP solution, and in the writing 350 uL/mL? Check and correct errors.

 L157-158: "...in agreement with our data, caffeic acid was reported as the most abundant phenolic acid.": taking into account the classification made in Table 2, if it is the most abundant; however, p-coumaric acid is the most abundant (39.8%) and it is also a phenolic acid. They should correct the wording.

 L160-161: It is a discussion that is not related to this manuscript, since they do not describe what type of material was used (roots, stem, leaf, bud, flowers or aerial parts) and the conditions of the material (dry or fresh).

 L163: correct "Chloric acid" to 'Cichoric acid'.

 L168-169: The authors mention that the chemical composition of propolis varies according to many factors, so why does the one used in this research not describe any of the factors present during its production?

 L185-186: How can you ensure that there were no significant differences in terms of sensitivity to EAP, if the results of Table 3 do not show signs (literals or asterisks) related to the statistical analysis of the data. Check and correct.

 L209: Clarify if the concentration at 300 or 350 uL/mL was used.

In general, in section 3 the discussions should be improved by highlighting the effects of the results obtained.

L218-219: Specify if you requested that the Hydroalcoholic solution of Echinacea purpurea (L.) Moench. and propilis be prepared for you. In this case, you have to describe in what proportion or amount of each product was used for the formulation or, if the combination used in the research is already sold as a commercial product (specify trade name, identification number, batch and date of manufacture).

 L223-226: It is necessary to provide information on the initial amount of Echinacea purpurea (L.) Moench and propolis that were used in the hydroalcoholic solution; as well as the method and conditions of preparation. If they cannot provide this information, the study may not be replicable and may not have scientific validity.

 L229: "About 2 mL of each extract were placed inside a 7 mL...": My question is: if after the extraction process (section 4.2) they ended up with 15 mg of residue, how do they start from 2 mL (make it clear if the residue was rediluted and with what?) of each extract (how many extracts were there?). There is only talk that they started with a hydroalcoholic solution of Echinacea purpurea (L.) Moench and propolis. Unless it is 2 mL of the Hydroalcoholic solution of Echinacea purpurea (L.) Moench. and propolis. Improve writing.

 L240-241: They must specify the model and characteristics of the mass spectrometer (simple, triple quadrupole?)

 L249: You must specify in the materials section, the mark of the alkanes standards.

 L265: Specify the type of column used for the GC-FID.

 L282-283: Were those amounts used from the original hydroalcoholic solution? If the solution contains alcohol, there is already inhibition of microbial growth. Or are they from the extract, but in what and how much was it rediluted?

 L310-311: They must specify that most of the results were treated by descriptive statistics and in the cases that used inferential statistics (ANOVA) they must specify which was the independent variable used as a source of variation, they must also mention the level of significance used . Based on the results in Figure 1, the ideal mean comparison test is Dunnett's, not Tukey's. Check and correct the wording.

L313-315: It is not a conclusion, it is the objective (remove it from this section).

Author Response

Reviewer 2

Dear Reviewer,

Thank you for having revised thoroughly our manuscript.

We have modified it according to the comments and the suggestions received by you and the other reviewers. Below you will find our point-to-point reply.

It is a work that leaves many doubts regarding the conditions for obtaining the hydroalcoholic solution of Echinacea purpurea (L.) Moench. and propolis that it is very important to know them in detail in order to carry out the replication and scientific validation of the results presented. In addition, the place of origin (from Northern Italy) is the place of the company, they do not ensure that it is where the plants and the propolis were produced.

In accordance with the reviewer's comments, we have added the requested information (section 4.1).

As mentioned in the manuscript, there are many factors that can influence the chemical composition of each of the two products used; therefore, with the commercial product they used (which does not describe any of its characteristics) the reproducibility of the experiment cannot be guaranteed. They should have done the collection, handling and processing of the plant and obtaining the propolis, by themselves. Or ensure that the commercial production of the studied product will always be under the same characteristics, factors and conditions.

The hydroalcoholic solution under investigation is a commercial product which guarantees the same preparation methods and therefore its characteristics are preserved.

In addition, I make the following specific observations:

 L79-82: I suggest removing the numerical values that are in parentheses, since the values can be clearly seen in the nearby table. It's redundant. Apply the same criteria in writing the other results.

As suggested by the reviewer, we have removed the numerical values in parentheses.

 L99-100: Table 2 header "COMPONENT1" has superscript 1 and is not described in table footer. Please put its meaning. In addition, it is important to explain the basis for the classification made for organic acids and phenolic acids. For example, why does caffeic acid classify it as a phenolic acid and p-coumaric acid classify it as an organic acid? when both are considered as organic compounds and classified as very similar hydroxycinnamic acids.

The reviewer is right; the superscript 1 has been removed and moved next to the abbreviation EAP. Its meaning has been reported in the legend.

Vanillic, salicylic, p-coumaric, ferulic, isoferulic, cinnamic acid and its derivatives such as, 3-hydroxycinnamic acid and 3,4-dimethoxycinnamic, have been correctly classified as phenolic acids.

 L122-125: It is an error to ensure that a slight decrease in the bacterial load was noted from the first hour of exposure, when the statistical analyzes indicate that there was no significant difference (absence of asterisk) between the three treatments. Therefore, there is no decrease. Correct the wording.

We thank you for the observation. The sentences were corrected as reported in lines 124-126.

 L125-127: Se desconoce si esa reducción fue causada por los compuestos en la solución de EAP o el diluyente (¿tal vez alcohol?) utilizado para el extracto. Por eso es importante que definan claramente todo el procesamiento del extracto.

The processing of the extract has been reported in the text. As regards the antimicrobial effectiveness, it is known that the presence of ethanol could affect the bioactivity of the extract. Nevertheless, is mainly the use of ethanol as a solvent that allows to considerably extract the phenolic and bioactive compounds of Echinacea purpurea (L.) Moench. and propolis, determining the antimicrobial effectiveness of the extract. The studies of Eshani et al. (2015)*, Borges et al. (2020)** and BozkuÅŸ & DeÄŸer (2022)*** affirmed that the ethanol, compared to other solvents, determines a better extraction yield of all those bioactive compounds of propolis, but also of other vegetable matrices.

Moreover, being hydroalcoholic solutions commonly used, their antimicrobial activity is widely reported in literature.

Lastly, in this manuscript we decided to report the results of the EAP extract, nevertheless we tested many other hydroalcoholic extracts on the same strains, obtaining different results that indicated EAP solution as the most effective. Therefore, although considering the antimicrobial activity of the ethanolic fraction of the hydroalcoholic solution, a contribution of the bioactive molecules contained in the EAP extract has been clearly confirmed by the data obtained.

* Ehsani, M.; Amin Marashi, M.; Zabihi, E.; Issazadeh, M.; Khafri, S. A Comparison between Antibacterial Activity of Propolis and Aloe vera on Enterococcus faecalis (an In Vitro Study). Int. J. Mol. Cell. Med. 2013, 2, 110-116.

** Borges, A.; José, H.; Homem, V.; Simões, M. Comparison of Techniques and Solvents on the Antimicrobial and Antioxidant Potential of Extracts from Acacia dealbata and Olea europaea. Antibiotics 2020, 9, 48.

*** BozkuÅŸ, T. N.; DeÄŸer, O. Comparison of total phenolic contents and antioxidant activities of propolis in different solvents. Food and Health 2022, 8, 111-117.

 L129-131: Why are the concentrations different at the beginning of the caption (350 and 400 uL/mL) and at the end (300 and 400 uL/mL)? correct at which concentration the actual test was done (at 350 or 300 uL/mL).

 L141-146: Why do you mention in Table 4 a concentration of 300 uL/mL of EAP solution, and in the writing 350 uL/mL? Check and correct errors.

We apologize for the mistakes. The concentrations have been corrected in lines 149, 161 and in Table 4.

 L157-158: "...in agreement with our data, caffeic acid was reported as the most abundant phenolic acid.": taking into account the classification made in Table 2, if it is the most abundant; however, p-coumaric acid is the most abundant (39.8%) and it is also a phenolic acid. They should correct the wording.

In agreement with the reviewer, we have changed the sentence.

 L160-161: It is a discussion that is not related to this manuscript, since they do not describe what type of material was used (roots, stem, leaf, bud, flowers or aerial parts) and the conditions of the material (dry or fresh).

Considering the information related to the type of material used for the preparation of the hydroalcoholic solution, we have decided to maintain the reference. We hope the reviewer agrees.

 L163: correct "Chloric acid" to 'Cichoric acid'.

Done.

 L168-169: The authors mention that the chemical composition of propolis varies according to many factors, so why does the one used in this research not describe any of the factors present during its production?

Details regarding the propolis production have been inserted in the text, in lines 251-255.

 L185-186: How can you ensure that there were no significant differences in terms of sensitivity to EAP, if the results of Table 3 do not show signs (literals or asterisks) related to the statistical analysis of the data. Check and correct.

We apologize for the mistake. No statistical analysis was conducted on the MICs and MBCs results, therefore the sentence was corrected by removing the word “significant” in line 202.

 L209: Clarify if the concentration at 300 or 350 uL/mL was used.

We thank the reviewer for the suggestion. The sentence has been corrected.

In general, in section 3 the discussions should be improved by highlighting the effects of the results obtained.

The discussions were improved along the text and the effects of the results obtained were underlined in lines 233-238.

L218-219: Specify if you requested that the Hydroalcoholic solution of Echinacea purpurea (L.) Moench. and propilis be prepared for you. In this case, you have to describe in what proportion or amount of each product was used for the formulation or, if the combination used in the research is already sold as a commercial product (specify trade name, identification number, batch and date of manufacture).

The hydroalcoholic solution of Echinacea purpurea (L.) Moench. and propolis used in this study was a commercial product. All requested information concerning the production process has been added in the text (section 4.1).

 L223-226: It is necessary to provide information on the initial amount of Echinacea purpurea (L.) Moench and propolis that were used in the hydroalcoholic solution; as well as the method and conditions of preparation. If they cannot provide this information, the study may not be replicable and may not have scientific validity.

As requested by the reviewer, the quantities of E. purpurea aerial parts and propolis used in the hydroalcoholic solution have been reported in the text (section 4.1).

 L229: "About 2 mL of each extract were placed inside a 7 mL...": My question is: if after the extraction process (section 4.2) they ended up with 15 mg of residue, how do they start from 2 mL (make it clear if the residue was rediluted and with what?) of each extract (how many extracts were there?). There is only talk that they started with a hydroalcoholic solution of Echinacea purpurea (L.) Moench and propolis. Unless it is 2 mL of the Hydroalcoholic solution of Echinacea purpurea (L.) Moench. and propolis. Improve writing.

According to the reviewer, we have changed the sentence in the text. The analysis performed by SPME-GC-MS was carried out directly on the starting hydroalcoholic solution (2 mL).

Subsequently, the extraction process was carried out to remove the water content to then carry out the derivatization reaction.

To avoid any confusion and better clarify the sequence of the experimental steps, we thought it appropriate to move section 4.2 and integrate it with section 4.5.

 L240-241: They must specify the model and characteristics of the mass spectrometer (simple, triple quadrupole?)

As requested by the reviewer, this information has been added.

 L249: You must specify in the materials section, the mark of the alkanes standards.

Done.

 L265: Specify the type of column used for the GC-FID.

Done.

 L282-283: Were those amounts used from the original hydroalcoholic solution? If the solution contains alcohol, there is already inhibition of microbial growth. Or are they from the extract, but in what and how much was it rediluted?

The specified concentrations are referred to the original hydroalcoholic solutions. The present work was extracted from a collection of preliminary analyses on hydroalcoholic solutions obtained from different botanical species (data not shown). Among them, the EAP solution was chosen for its higher microbial effectiveness in terms of MICs. Therefore, as the extracts were prepared following the same extraction method and with the same solvent, the antimicrobial activity of EAP was attributed to its bioactive compounds. The influence of ethanol on the antimicrobial effectiveness of the EAP solution was discussed in the previous question regarding lines 125-127.

 L310-311: They must specify that most of the results were treated by descriptive statistics and in the cases that used inferential statistics (ANOVA) they must specify which was the independent variable used as a source of variation, they must also mention the level of significance used. Based on the results in Figure 1, the ideal mean comparison test is Dunnett's, not Tukey's. Check and correct the wording.

We thank the reviewer for the suggestion. According to the observation, the sentences were modified as reported in lines 350-354.

L313-315: It is not a conclusion, it is the objective (remove it from this section).

As suggested by the reviewer, the lines have been removed from the text.

Round 2

Reviewer 2 Report

I appreciate that you have followed the recommendations made. The manuscript was improved and most of the doubts were clarified.